# Preparation, Characterization, and Terahertz Spectroscopy Characteristics of Reduced Graphene Oxide-Doped Epoxy Resin Coating

**Mian Zhong** [1][ID]**, Xin Dai** [1]**, Hongxing Xiang** [1]**, Bingwei Liu** [2][ID]**, Xin Zhao** [1,*]**, Dongshan Wei** [3,*]**, Xiaoguang Tu** [1]**, Zhihao Wang** [4]**, Yan Gong** [5]**, Yajun Xu** [1] **and Mingxia He** [4,*]

1  Institute of Electronic and Electrical Engineering, Civil Aviation Flight University of China, Guanghan, Deyang 618307, China; zhongmian182@vip.163.com (M.Z.); dxcafuc@163.com (X.D.); x87729637@gmail.com (H.X.); xguangtu@outlook.com (X.T.); genius98@126.com (Y.X.)
2  School of Optical-Electrical and Computer Engineering, University of Shanghai for Science and Technology, Shanghai 200093, China; bingweiliu_work@126.com
3  School of Electrical Engineering and Intelligentization, Dongguan University of Technology, Dongguan 523808, China
4  School of Precision Instrument and Opto-Electronics Engineering, Tianjin University, Tianjin 300072, China; wangzhihao199987@163.com
5  School of Materials Science & Engineering, Beijing Institute of Fashion Technology, Beijing 100029, China; clygy@bift.edu.cn
*  Correspondence: zhaox@cafuc.edu.cn (X.Z.); dswei@dgut.edu.cn (D.W.); hhmmxx@tju.edu.cn (M.H.)

**Abstract:** Reduced graphene oxide has attracted numerous interests due to its unique, superior electronic, optical, mechanical, and chemical properties. An epoxy resin with excellent mechanical and electrical properties can be obtained by doping with reduced graphene oxide to enhance the function of the polymer. Here, we prepared a uniform reduced graphene oxide/epoxy resin coating with a different reduced graphene oxide content and characterized it using a field-emission scanning electron microscope (FE-SEM), X-ray diffractometer (XRD), Raman, and Fourier transform infrared spectrometer (FTIR). Furthermore, the spectral characteristics of the composite coating in the terahertz band were discussed. The cross-sectional SEM results show that a fold structure with ductile failure was intensively formed due to the compatibility of graphene and polymer materials. Both the Raman G and Raman 2D peaks of reduced graphene oxide were confirmed using Raman spectrum testing. The diffraction peak of reduced graphene oxide at 24° disappeared within the reduced graphene oxide/epoxy resin coating, and a wide diffraction peak of the amorphous structure was formed together. Additionally, the intensity of the Raman spectrum increased significantly with increased reduced graphene oxide content, thereby making the surface electrical resistance of the coatings decrease exponentially. Additionally, the intensity of the terahertz time-domain signal and frequency-domain power spectrum linearly reduced with increased reduced graphene oxide concentration. However, the terahertz absorption coefficient and refractive index both increased gradually with increased reduced graphene oxide doping due to increased orientation polarization in the composite coating.

**Keywords:** terahertz spectroscopy; XRD; FE-SEM; Raman; FTIR

## 1. Introduction

Reduced graphene oxide (rGO) has attracted widespread interest in the previous decade due to its unique and excellent properties, such as high surface area, good elasticity, thermal stability, and electrical conductivity [1–3]. rGO is considered a promising additive material for improving the mechanical, physical, chemical, and thermal stability of substrate coatings (e.g., strength, modulus, friction, adhesion, and wear resistance) [4–7].

Epoxy resin (EP) is widely used as functional coatings for civil aircraft due to its excellent corrosion resistance, mechanical properties, and chemical stability. Reports show that graphene-based resin coating can be employed for anticorrosion, antierosion, electromagnetic interference shielding, and other protection [8–14]. For instance, the wear resistance property and corrosion protection of EP can be remarkably enhanced by incorporating graphene [15]. For example, the graphene film with a network structure coated on the surface of the resin matrix or aircraft parts has been applied for deicing the electrothermal effect after heating [16].

Terahertz time-domain spectroscopy (THz-TDS), as a nondestructive technique, has been applied to qualitatively and quantitatively detect surface defects of aeronautical composite materials through spectral characteristic analyses of the aeronautical composite matrix resins by measuring the absorption, transmission, refraction, and dielectric dispersion at the THz frequency range [17–21]. However, the THz spectral characteristics of internal impurities, especially for carbon additives, in composite materials, are affected by various factors, such as polarization directions and/or electrical conductivity, type and amount of filler, geometry, and dispersibility of additives [22–24]. However, the amount of filler is a key factor. Therefore, it is necessary to investigate the effects of different doping rGO contents on THz spectroscopy characteristics.

In this paper, we mainly focused on the physical and chemical characterization of rGO/EP composite coating filled with different concentrations of rGO additives. Furthermore, we investigated various THz spectral characteristics, thereby obtaining the relationship between THz parameters and rGO additives. Finally, we explained the THz absorption enhancement mechanism of rGO/EP composites.

## 2. Experimental Section

### 2.1. Materials

Pristine reduced graphene oxide (rGO) was purchased from Hangzhou Hangdan Optoelectronic Technology Co. Ltd. (Hangzhou, China). It has few layers with a thickness of 0.7 nm and a horizontal size ranging from 0.5 to 10 μm for each layer, and the purity is over 99%. Additionally, sodium dodecylbenzene sulfonate (SDBS) was provided by Sinopharm Chemical Reagent Co. Ltd. (Shanghai, China). Then, EP and the curing agent were supplied by Changzhou Runxiang Chemical Co. Ltd. (Changzhou, China).

### 2.2. Preparation of rGO/EP

Figure 1 shows the preparation process for rGO/EP coating. The rGO was immersed in an anhydrous ethanol solution for 8 h with a small amount of SDBS added as surfactant. Then, the solution was treated with mechanical stirring and ultrasonic oscillating for at least 10 min to disperse the additives. Then, a certain amount and proportion of EP and curing agents, respectively, were subsequently added. After stirring the mixed solution evenly, it was continuously vacuumed for 3 min at 0.52 standard atmospheric pressure and repeated 5 times to minimize the influence of bubbles in the mixed system. Next, the composite coating spread on a quartz glass was repeatedly coated using a 4-sided applicator (TQC sheen, Capelle aan den IJssel, The Netherlands) until its surface was smooth and uniform. Finally, the coated sample was placed in a drying oven at a constant temperature of 60 °C for 3 h until the film was completely cured. Using this process, we prepared the rGO/EP composite samples for different rGO doping concentrations (1.0 wt.%, 3.0 wt.%, and 5.0 wt.%), including the one without rGO doping (0.0 wt.%).

### 2.3. Characterization

An FEI quanta 250FEG field-emission scanning electron microscope (FE-SEM, FEI, Hillsboro, OR, USA) was used to observe the cross-section morphology of the rGO/EP coating. The physical and chemical characterizations of the rGO/EP coating were investigated using a Nicolet iS5/10 Fourier-transform infrared spectrometer (FTIR, Thermo Fisher Scientific, Waltham, MA, USA), TD-3500 X-ray diffractometer (XRD, Dandong Tongda

Science & Technology Co., Ltd., Dandong, China), and Renishaw inVia Raman spectrometer (Renishaw, London, UK). An MS5205 insulation resistance tester (Peakmeter, Co., Ltd., Shenzhen, China) was used to test the surface electrical resistance of the rGO/EP coating. Furthermore, THz spectral characterization of the composite coating samples was performed using an Advantest TAS7400 terahertz spectroscopy system (Advantest, Tokyo, Japan).

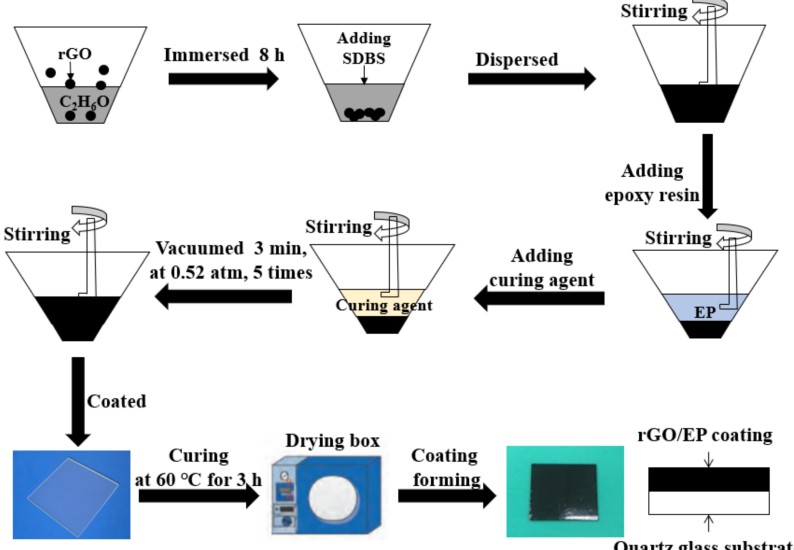

**Figure 1.** Preparation flowchart of the rGO/EP coating.

## 3. Results and Discussion

### 3.1. Morphology Analysis

Figure 2 shows the cross-section morphology of rGO/EP coatings with different rGO dopings characterized using SEM. In Figure 2a, the pure EP coating has a flat cross section without dispersive fillers and is a linear propagating crack, which is one characteristic of a typical brittle fracture of the thermosetting resin. The cross-section morphology of rGO/EP completely changes with increased rGO content, as shown in Figure 2b–d. Additionally, folded structures, which are related to the two-dimensional rGO network structure, were extensively formed. The folded structures show the characteristics of ductile failure, indicating that the toughness of pure EP is greatly changed by adding rGO micro-/nanomaterials. Due to the compatibility of rGO and polymer materials, an ideal interface can be formed between both materials.

### 3.2. Physical and Chemical Characterizations

Figure 3 shows the physical and chemical characterizations of rGO/EP coatings with different rGO doping concentrations. Two rGO Raman characteristics were observed, namely, the G and 2D peaks located at 1580 and 2930 cm$^{-1}$, respectively, as shown in Figure 3a. However, no obvious peak exists in the pure EP coating, and its Raman spectrum intensity is significantly higher than those of rGO/EP coatings. Results show that the corresponding G peak near 1580 cm$^{-1}$ is related to the ordered sp2 carbon atom vibration in rGO, which is an intrinsic Raman mode of rGO. Moreover, an increased doping concentration decreases the Raman spectral intensity at the same Raman frequency position.

Additionally, in the XRD curves in Figure 3b, there is a weak wide diffraction peak, centered at 24° of the rGO additives, ranging from 10° to 40°. This peak is mainly caused by the 002-layer graphite structure, which is created by the accumulation of rGO. Additionally, the coating film of the pure EP system has a strong wide diffraction peak, centered at 22°, from 10° to 30°. This is due to the diffraction peak reflected in the random amorphous phase region in the curing system of the EP polymer. However, for rGO/EP composite coatings, we can see that the diffraction peak of rGO at 24° disappeared, indicating that the

rGO in the peel state was uniformly dispersed in the EP matrix. Besides, the dispersion of rGO layers within the EP matrix can be measured using Raman spectroscopy. Another interesting observation is that there is a new and strong narrow diffraction peak of the rGO/EP, centered at 19°. This suggests that the rGO additives are bound to the EP matrix in a new form. The intensity of this peak increases significantly with increased rGO content, which is mainly correlated with the accumulation of the rGO in the polymer matrix.

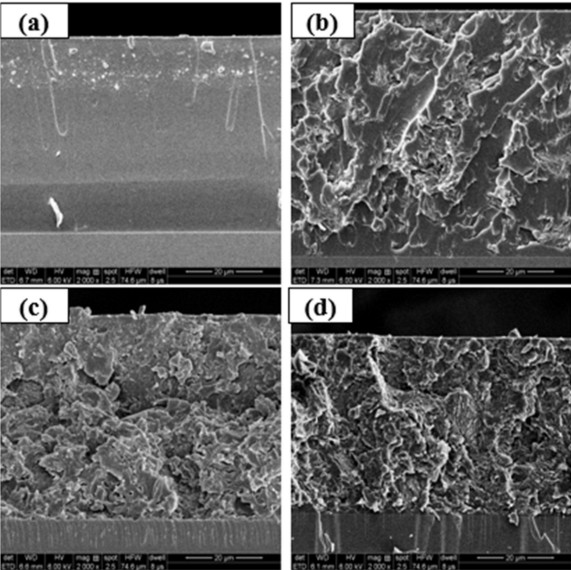

**Figure 2.** Cross-section SEM images of EP coatings with different rGO doping concentrations: (**a**) pure, (**b**) 1.0 wt.%, (**c**) 3.0 wt.%, (**d**) 5.0 wt.%.

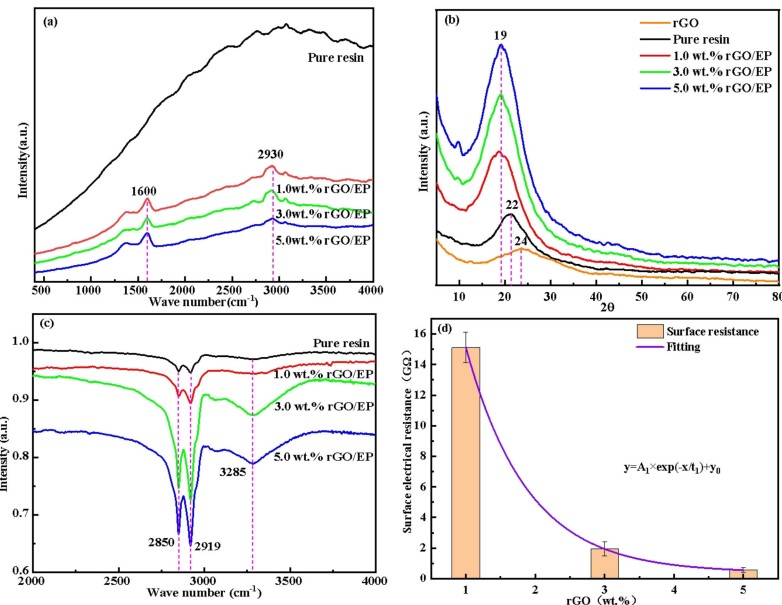

**Figure 3.** Characterization of rGO/EP coatings: (**a**) Raman, (**b**) XRD, (**c**) FTIR, (**d**) surface electrical resistance.

The performance of polymer composite coatings is determined by the interaction between the modified dopants and the substrate. Figure 3c shows the stretching vibration peaks of –CH₂– located at 2919 and 2850 cm⁻¹. The amplitude of the N–H characteristic peak located at 3285 cm⁻¹ gradually increases with increased rGO content, indicating that the hydrogen bond between the conductive filler and matrix forms becomes stronger. Especially, the IR spectrum of the rGO/EP at 5 wt.% presents a strong absorption peak;

consequently, an absorption peak shift is observed in the IR spectra, which is caused by the strong interaction between rGO mixtures [25]. Additionally, compared with the pure EP coating, the redshift in the rGO/EP composite coating indicates that the vibration energy of the functional groups of the composite system is smaller and more stable. This is caused by hydrogen bonds formed between the residual oxygen-containing groups on the surface of rGO and functional groups in the polymer, consequently making the composite system more stable.

The pure EP system is a good insulator. Experimental results show that adding rGO with good conductivity essentially influences the surface electrical resistance of the rGO/EP composite coatings. Consequently, the surface electrical resistance of the coatings decreases exponentially with increased rGO additive content. In other words, the rGO/EP coating exhibits a relatively higher electrical conductivity due to the formed rGO conductive networks.

### 3.3. Terahertz Spectroscopy Testing

A series of rGO/EP composite coatings with different doping contents were prepared for THz spectroscopy measurements. Figure 4a is the THz time-domain spectra of rGO/EP coatings with different doping concentrations. The signal intensity of THz electromagnetic radiation after passing through the epoxy material linearly decreases with the increase in the conductive rGO concentration, as shown in Figure 4a. This is because of the strong absorption of THz waves by an electric field, produced using the directional movement of free electrons that occur on the sample coating surface.

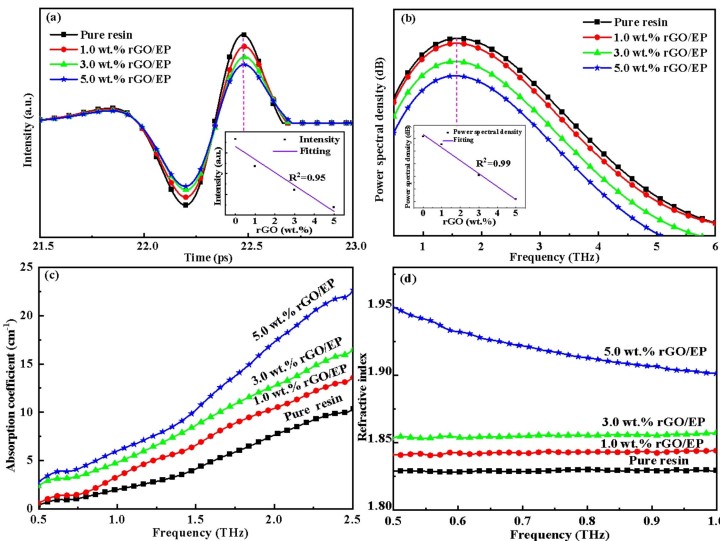

**Figure 4.** Terahertz spectral characterization of rGO/EP coatings: (**a**) time domain, (**b**) frequency domain, (**c**) absorption coefficient, (**d**) refractive index.

The frequency-domain spectra were obtained using the Fourier transform of the time-domain spectral data, as shown in Figure 4b. The spectral results show that the power spectral density of these four samples increased first and then decreased with increased frequency in the range of 0.5–6.0 THz. Additionally, the intensity of the power spectral density gradually decreased with increased rGO doping concentration, which is mainly due to the dipole polarization in the composite coatings. As the frequency increases, especially in the high-frequency band (1.5–6 THz), the dipole orientation polarization speed fails to match the changing speed of the electromagnetic field. Therefore, the dipole orientation polarization needs to consume more energy to overcome the internal resistance, which increases the loss of the power spectral density. To further study the relationship between the power spectral density at the maximum peak frequency and the rGO doping concentration, as shown in the inset in Figure 4b, we found that the power spectral density is linear with the concentration due to the increased loss in dipole orientation polarization

caused by the interaction of micro/nanographene and polymer long chains (e.g., tangles and/or clusters). Figure 4c shows the relationship between the absorption coefficient and THz frequency. The absorption coefficient increases with increasing frequency ranging from 0.5 to 2.5 THz and with increased rGO concentration. Figure 4d illustrates the variation of the refractive index of rGO/EP with different rGO doping concentrations ranging from 0.5 to 1.0 THz. The refractive index of the composite coating increases with increased rGO concentration. Compared with composite coatings with low concentration ($\leq$3%), the refractive index of rGO/EP tends to be saturated with the frequency. However, when the rGO doping concentration is high (5%), the refractive index of the composite coating is significantly higher than those at low concentrations and remarkably decreased with increased frequency.

For the ultrahigh frequency THz band, the polar functional group contained in the composite coatings causes dipole orientation polarization. When the THz frequency increases, the polarization orientation speed of the dipole becomes faster, which results in a loss of directional polarization, manifested as a violent relaxation motion. Therefore, as the frequency increases, the absorption coefficient increases. Additionally, under the THz electric field, the relative displacement between charged carriers in graphene with excellent conductivity is approximately equivalent to the dipole relaxation, and the absorption of THz by carriers comes from the equivalent dipole. An increase in rGO concentration increases the number of carriers and the dipole relaxation phenomena. Consequently, the absorption coefficient increases with the increase in rGO concentration.

## 4. Conclusions

In this paper, we investigated the effect of nanographene conductive additives on the morphology, electrical conductivity, and terahertz spectroscopy of the EP matrix. Results show that the nanographene conductive additives dispersed into the EP matrix and formed a good combination due to the compatibility of both materials. Additionally, the electrical conductivity of rGO/EP closely depends on the number of nanographene additives. With an increase in the added content, the electric field intensity decreased successively, corresponding to enhancing the THz wave absorption by the additives. Optical parameters, such as terahertz absorption coefficient and refractive index, were related to the electrical conductivity and distribution of doped nanoadditives. It is found that the introduction of rGO effectively enhanced the THz band absorption of the EP matrix. This study provides a certain reference not only for further research on the dielectric response mechanism of nanoconductive particles under the interaction of the THz radiation field but also for concentration detection and quantitative analysis using THz spectral detection due to its high sensitivity to changes of the conductive additives in composite materials.

**Author Contributions:** Conceptualization, data curation, funding acquisition, resources, and writing—original draft preparation, M.Z.; data curation and writing—original draft preparation, X.D.; data curation and resources, H.X.; conceptualization and resources, B.L.; funding acquisition and project administration, X.Z.; supervision and writing—review and editing, D.W.; data curation and software, X.T.; resources and software, Z.W.; writing—review and editing, Y.G.; funding acquisition, Y.X.; supervision and writing—review and editing, M.H. All authors have read and agreed to the published version of the manuscript.

**Funding:** The authors gratefully acknowledge the financial support from the National Key R&D Program of China (Grant No. 2018YFC0809500), Talents Program of Civil Aviation Education (Grant No. 0252123), and Project of Civil Aviation Flight University of China (Grant Nos. J2018-56, J2020-49, and CJ2019-01).

**Institutional Review Board Statement:** Not applicable.

**Informed Consent Statement:** Not applicable.

**Data Availability Statement:** Data sharing is not applicable to this article.

**Conflicts of Interest:** The authors declare no conflict of interest.

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
