# Peer review of "Preparation, Characterization, and Terahertz Spectroscopy Characteristics of Reduced Graphene Oxide-Doped Epoxy Resin Coating"

_coatings, doi:10.3390/coatings11121503_

Round 1
Reviewer 1 Report
The manuscript Deals with the obtention of an epoxy resin (EP) with excellent mechanical and electrical properties by doping with reduced graphene oxide to enhance its function. And its characterization using field emission scanning electron microscopy(FESEM), X-Ray diffractometer(XRD), Raman, and Fourier transform infrared spectrometer(FTIR).
It seems the manuscript has been all in all, well mapped out and understandably presented, and could be interesting for the readership working on the topic or parallelly closer to. I placed all my annotations, suggestions and questions/doubts throughout the manuscript, to be checked thoroughly by the authors.

Author Response
Response to Reviewer’s comments(coatings-1484135)
Reviewer1:
The manuscript Deals with the obtention of an epoxy resin (EP) with excellent mechanical and electrical properties by doping with reduced graphene oxide to enhance its function. And its characterization using field emission scanning electron microscopy(FESEM), X-Ray diffractometer(XRD), Raman, and Fourier transform infrared spectrometer(FTIR).
It seems the manuscript has been all in all, well mapped out and understandably presented, and could be interesting for the readership working on the topic or parallelly closer to. I placed all my annotations, suggestions and questions/doubts throughout the manuscript, to be checked thoroughly by the authors.
- Having been introduced what these acronyms stand for in the introduction, it's better to avoid them on the abstract.
We thank the reviewer’s interest and positive comments on our work. According to the reviewer’s valuable suggestions, we have carefully revised our manuscript thoroughly. The acronym error has been fixed in the introduction section.
- Perhaps I ain't catch correctly the authors drift - does Pristine-reduced graphene oxide means, rGO with few layers such that the thickness of each 66layer is 0.7 nm, horizontal size of 0.5–10 μm, and purity >99%?
In this work, rGO is the acronym of pristine-reduced Graphene Oxide, which only contains few layers with the thickness of 0.7 nm and the horizontal size ranging from 0.5 to 10 μm for each layer, and the purity is over 99%. We have rephrased this sentence in the revised version (Line 70-73).
- In my opinion it would have sounded better if was stated, "...we prepared thee rGO/EP composite samples for different rGO doping concentration ( 1.0 wt.%, 3.0 wt.%, and 5.0 wt.%), including the one without rGO doping (0.0 wt. %). See thecomment on figure.
Thanks very much for the reviewer’s useful suggestions. We have modified this sentence in the new version (Line 91-93).
- The pure state is it for the epoxy resin, correct? I think would be better stating,...EP coating with different rGO doping concentrations. The EP resin (epoxide+polyamine) at pure state do not conform rGO/EP yet.
Thanks very much for the reviewer’s useful suggestions. We have modified this sentence in the version (Line 107-108).
- What kind of surface resistance, stiffness, hardness or electrical resistance?
In this work, we mainly focused on the surface electrical resistance of the rGO/EP composite coatings.
- What CFRP stands for?
“CFRP” means carbon fiber reinforced polymer composite. It should be EP rather than CFRP in this paper. We are sorry for this typo.
- According to the figure 4 (d) the behaviour of the curve for 5% wt decreases with the increasing frequency. It seems a bit opposite to the statement, isn't it?
We apologized that we did not explain this clearly in the submitted manuscript. As seen from Figure 4(d), the refractive index of the composite coating increases with increased rGO concentration, however, the refractive index of the composite coating decreases with increased frequency only for the case of the rGO doping concentration is high (≥5%). We have made this statement clear in the new version.
- This section has to do with THz band, therefore, the ultra-high frequency band is inner THz frequency band.
The reviewer is correct. The ultra-high frequency band is inner THz frequency band. We have modified this sentence in Line 204.
- Have the curing elements some influence/interference on the EP resin and rGO/EP composites characterization results?
Thanks very much for the reviewer’s valuable suggestions. Actually, the curing elements have an effect on characterization results of the EP resin and rGO/EP composites. However, for simplicity, the curing elements have not been discussed in this paper. We believe that it is worth of further studies.
Other modifications
- Figures 1-4 have been changed.
- The phrase of surface resistance has been changed to surface electrical resistance.
- The two references as of Refs. 13 and 14 have been added.
- The original references 13-22 have replaced by the new references 15-24.
- A new reference (No.25) has been added.

Reviewer 2 Report
In this paper, the authors reported the preparation of uniform reduced graphene oxide / epoxy resin (rGO/EP) coating with different concentration of rGO.
The characterization of the composite coating rGO/EP was performed suitably by using various methods such as SEM, XRD, Raman spectrum, FTIR, and terahertz time-domain spectroscopy.
I think that the results reported in this paper are useful to develop the field of composite coating.
There are some comments as follows.
- You may mention more clearly the advantage and disadvantage of the composite coating rGO/EP comparing with original epoxy resin.
- Line 127, fig. 3 (b)
What do you think about the peak at 19°?
- Line 135
The redshift in the rGO/EP is not clear.
- 3, fig. 4
GO/CFRP → GO/EP ?
Author Response
Response to Reviewer’s comments(coatings-1484135)
Reviewer2:
In this paper, the authors reported the preparation of uniform reduced graphene oxide / epoxy resin (rGO/EP) coating with different concentration of rGO.The characterization of the composite coating rGO/EP was performed suitably by using various methods such as SEM, XRD, Raman spectrum, FTIR, and terahertz time-domain spectroscopy.
I think that the results reported in this paper are useful to develop the field of composite coating.
There are some comments as follows.
- You may mention more clearly the advantage and disadvantage of the composite coating rGO/EP comparing with original epoxy resin.
We appreciate the reviewer’s good suggestions. We have added two references to support rGO/EP for better electromagnetic interference shielding performance (Refs. 13 and 14 in the revised manuscript). In this paper, the experimental results show that the introduction of rGO effectively enhanced the THz wave absorption of the EP matrix (Line 225-226 ). Therefore, rGO/EP is more suitable for applications in the THz frequency bands.
- Line 127, fig. 3 (b)What do you think about the peak at 19°?
Thanks for the reviewer’s valuable suggestions. We have added the corresponding description of the peak at 19°in the revised manuscript(Line 142-146).
- Line 135The redshift in the rGO/EP is not clear.
We thank the reviewer for pointing out this issue. The redshift in the rGO/EP has been clearly described in Line 152-154 in the revised version.
- Fig 3, fig. 4GO/CFRP → GO/EP ?
We apologized for this typo. We have fixed this error in the new version. Figure 3 and Figure 4 have also been replotted.
Other modifications
- Figures 1-4 have been changed.
- The phrase of surface resistance has been changed to surface electrical resistance.
- The two references as of Refs. 13 and 14 have been added.
- The original references 13-22 have replaced by the new references 15-24.
- A new reference (No.25) has been added.

Reviewer 3 Report
The work "Preparation, characterization and terahertz spectroscopy characteristics of reduced graphene oxide doped epoxy resin coating" is interesting and present useful results.
The structure of the manuscript is good and the subject is very actual for the research field.
The procedures and methods were well presented.
The results are interesting and clearly presented.
The reference covering the specific field of the research, but please mention all co-authors (generally, the reference must be write at journal standards).
However, presentation of the surfaces SEM images of rGO/EP coatings with different rGO doping concentrations: 98(a) pure; (b) 1%; (c)3%; (d)5% were welcome.
Author Response
Response to Reviewer’s comments(coatings-1484135)
Reviewer3:
The work "Preparation, characterization and terahertz spectroscopy characteristics of reduced graphene oxide doped epoxy resin coating" is interesting and present useful results.The structure of the manuscript is good and the subject is very actual for the research field.The procedures and methods were well presented.The results are interesting and clearly presented.
The reference covering the specific field of the research, but please mention all co-authors (generally, the reference must be write at journal standards).
However, presentation of the surfaces SEM images of rGO/EP coatings with different rGO doping concentrations: 98(a) pure; (b) 1%; (c)3%; (d)5% were welcome.
We thank the reviewer’s positive comments and high evaluation on our work. The reference format is modified to meet the journal’s requirement and three new references have been added in the revised version.
We appreciate the reviewer’s good suggestions. As for the cross-section SEM images of EP coatings with different rGO doping concentrations (shown in Figure 2), it was found that the cross-section morphology of rGO/EP completely changes with increased rGO content. Additionally, the folded structures show that the toughness of pure EP was greatly changed by adding rGO micro/nano-materials. Therefore, we think the cross-section SEM is enough to support the evolution of the surface morphology of the rGO/EP to a certain degree.

Round 2
Reviewer 2 Report
I think that the revised manuscript is much better than the original one.